# Evaluation of Urine and Vaginal Self-Sampling versus Clinician-Based Sampling for Cervical Cancer Screening: A Field Comparison of the Acceptability of Three Sampling Tests in a Rural Community of Cuenca, Ecuador

**DOI:** 10.3390/healthcare10091614

**Published:** 2022-08-25

**Authors:** Bernardo Vega Crespo, Vivian Alejandra Neira, José Ortíz S, Ruth Maldonado-Rengel, Diana López, Andrea Gómez, María José Vicuña, Jorge Mejía, Ina Benoy, Tesifón Parrón Carreño, Veronique Verhoeven

**Affiliations:** 1Facultad de Ciencias Médicas, Universidad de Cuenca, Cuenca 010203, Ecuador; 2Facultad de Medicina, Universidad del Azuay UDA, Cuenca 010104, Ecuador; 3Facultad de Ciencias de la Salud, Universidad Técnica Particular de Loja UTPL Loja Ecuador, Loja 1101608, Ecuador; 4Programa de Doctorado en Ciencias Morfológicas, Universidad de La Frontera UFRO, Temuco 4811230, Chile; 5AMBIOR, Laboratory for Cell Biology and Histology, University of Antwerp, 2610 Antwerp, Belgium; 6Facultad de Ciencias de la Salud y Neurociencias, Universidad de Almería UAL, 04120 Almería, Spain; 7Family Medicine and Population Health, University of Antwerp, 2610 Antwerp, Belgium

**Keywords:** HPV, self-sampling, urine sampling, vaginal sampling, clinician sampling, acceptability

## Abstract

Self-sampling methods for HPV testing have been demonstrated to be highly sensitive and specific. The implementation of these methods in settings with a lack of infrastructure or medical attention has been shown to increase the coverage of cervical cancer screening and detect cervical abnormalities in the early stages. The aim of this study is to compare the acceptability of urine and vaginal self-sampling methods versus clinician sampling among rural women. A total of 120 women participated. Each participant self-collected urine and vaginal samples and underwent clinician sampling for Pap smear and HPV testing. After the sample collection, a questionnaire to qualify the device, technique, and individual acceptability was applied, and the additional overall preference of three sample tests was evaluated. Results: The characteristics of the participants were as follows: median age of 35 years; 40.8% were married; 46.7% had a primary level of education; median age of sexual onset of 17.6 years. Compared with clinician sampling, both vaginal self-sampling, OR 20.12 (7.67–52.8), and urine sampling, OR 16.63 (6.79–40.72), were more comfortable; granted more privacy: vaginal self-sampling, OR 8.07 (3.44–18.93), and urine sampling, OR 19.5 (5.83–65.21); were less painful: vaginal self-sampling, OR 0.07 (0.03–0.16), and urine sampling, OR 0.01 (0–0.06); were less difficult to apply: vaginal self-sampling, OR 0.16 (0.07–0.34), and urine sampling, OR 0.05 (0.01–0.17). The overall preference has shown an advantage for vaginal self-sampling, OR 4.97 (2.71–9.12). No statistically significant preference was demonstrated with urine self-sampling versus clinician sampling. Conclusions: Self-sampling methods have a high acceptance in rural communities. Doubts on the reliability of self-sampling often appear to be a limitation on its acceptability. However, the training and education of the community could increase the uptake of these methods.

## 1. Introduction

Cervical cancer (CC) could be the first cancer eliminated in the world [1,2], through effective strategies such as 90-70-90, proposed by the World Health Organization (WHO) in 2020. This initiative proposes that 90% of women should be vaccinated against human papillomavirus (HPV); 70% of all women must be screened with a high-sensitivity test at least twice during their lifetime; 90% of women identified with a cervical disease must receive treatment [3]. This promising horizon is a goal for many countries, such as Australia, where the deadline to eliminate cervical cancer is set at 2035 [4]. In addition to vaccination, the use of HPV detection in primary screening for CC is an important tool to reach this objective; these types of tests have shown high effectiveness compared with Pap smears in the detection of cervical abnormalities [5,6]. Despite these advances, CC is still a threat to women’s lives. In 2020, CC caused 341.831 early deaths worldwide [7]. Of all the decreases, 80% occurred in low- and middle-income countries (LMICs) [8].

In South America, Bolivia has the highest rate of CC mortality (35.8/100.000); Brazil has the lowest incidence rate (12.2/100.000). The incidence rate of CC in Ecuador has reached 17.8/100.000 [9].

The rates of CC screening are closely related to mortality, the former of which are lower in LMICs [10]. The low coverage of CC screening is an important problem in Ecuador, where 41.6% of women of reproductive age have never been screened [11]. The main associations related to low screening coverage are availability, acceptability, and accessibility to quality services [12,13]. In addition, women consider clinician sampling as intrusive and painful experiences; these considerations could reduce the CC screening uptake [14]. In addition, people with a lack of knowledge on cervical cancer prevention could have less adherence to screening due to the lack of risk perception as well as embarrassment [13,15].

Self-sampling methods, such as vaginal self-sampling, have an emerging role in overcoming barriers to cervical cancer screening. The acceptability among women varies from 82% to 93.4% for vaginal self-sampling [16,17] and 94% to 95% for urine self-sampling [17,18,19]. The reliability for the diagnosis of HPV of self-sampling tests is adequate: the sensitivity ranges from 84.6% to 94.4% and specificity from 92.1% to 98% for vaginal self-sampling [20,21,22], and the sensitivity ranges from 88.8 to 90.5% and specificity from 828% to 94.1 for urine self-sampling [20,23,24]. Self-sampling tests reduce needs in terms of infrastructure [17], and it has been demonstrated that self-sampling methods are optimal in terms of the cost-effectiveness [25].

Despite the evidence of the acceptability and reliability of self-sampling methods, there is still a limited number of studies in Latin America on these techniques and their acceptability. The literature reveals that the main barriers to the acceptance of self-sampling methods are concerns about not performing the test correctly, hurting oneself, and the accuracy of the test [26,27].

In Ecuador, self-sampling methods for HPV testing are not yet implemented, and this is the first study that has been conducted in the country to assess to what extent self-sampling methods would be accepted by women.

In rural communities in Ecuador, access to screening is low, and women in the local population have mixed cultural roots. Cultural beliefs possibly influence screening preferences. Therefore, the aim of this study was to compare the acceptability of urine and vaginal self-sampling methods versus clinician sampling among rural women.

## 2. Materials and Methods

***Ethical statement:*** This study was approved under the guidance of the Declaration of Helsinki and the Council for International Organizations of Medical Sciences (CIOMS). All procedures involving human participants were approved by the bioethical committee of the University of Cuenca (approval code: UC-COBIAS-2020-262) and the National Directory of Intelligence in Health (DIS) of the Ministry of Health of Ecuador (approval code: MSP-DIS-2020-0405-O). All of the participants were informed about the purpose of the study and signed informed consent before the sample collection.

***Study population:*** A diagnostic test study was conducted in the rural parish of El Valle of Cuenca, Ecuador, from May to August 2021. Through flyers delivered in public places, households, and healthcare centers, the women of El Valle were invited to participate.

The inclusion criteria included being sexually active; being between 18 and 70 years old; not having undergone an excision or destructive treatment of cervical intraepithelial neoplasia; not having used vaginal medication at least a week before the examination; not having had sexual intercourse for at least 48 h previous to the examination; not being pregnant; and the absence of menstrual bleeding at the examination.

### 2.1. Sample Collection

Prior to the sample collection, women who agreed to participate were taught how to take the samples: a pictographic representation of each technique was given to all of the participants, and the same graphics were present in the bathroom where the participants obtained the samples.

The first sample obtained was a urine sample: it was obtained directly in a sterile urine container after self-made asepsis in the bathroom of the consultation room. All of the participants were asked to collect at least 30 cc of urine. After urine sampling, a self-sampling device for vaginal sampling was given to the patients. An Evalyn^®^ Brush from Rovers Medical Devices was the selected tool for this sampling. Fabricant instructions were used for the sample collection; the researchers waited outside of the bathroom for any additional explanation and to receive the samples after their collection. Finally, the patients were subjected to a clinician-acquired sample in the standard manner. After the insertion of the speculum, endocervix and exocervix samples were obtained by using a Hybribio cervical brush ^®^, rotating 360° twice.

### 2.2. Evaluation of Acceptability

All of the participants that underwent urine self-sampling, vaginal self-sampling, and clinician sampling were included in the study. The results of the test properties have been reported in https://www.mdpi.com/1660-4601/19/8/4619/htm (accessed on 1 July 2022).

After the clinician sampling, participants were asked to fill in a questionnaire. The questionnaire was based on a literature review and on the one applied by Shin et al. in Korea in 2019 [17]. The final questionnaire was reviewed and validated by local experts.

For the measurement of acceptability, three categories were identified: 1. Device and technique acceptability. The questions included were as follows: Which test is faster to be used? Which test do you find more comfortable? Which test do you think is reliable for diagnosis? Which test do you find easier to use? Which test gives you more privacy? Which test gives you more confidence in taking a sample? A Likert scale was used to qualify the tests (1: very dissatisfied, 2: dissatisfied, 3: indifferent, 4: satisfied, and 5: very satisfied). 2. Individual perception of the test. The questions included were as follows: Level of embarrassment caused by the method. Level of mistrust that the method causes you. Level of pain that the method causes you. Level of difficulty in the application of the method. A Likert scale was used to qualify the acceptability (1: very dissatisfied, 2: dissatisfied, 3: indifferent, 4: satisfied, and 5: very satisfied). Finally, an overall preference for the method was determined (1: I would not choose it again, 2: indifferent, and 3: I would always choose it).

Likert scales were used due to their proprieties allowing for the evaluation of less concrete concepts, as well as the fact that they help to estimate how much of the variability within each measure is due to factors such as participant-level individual differences and item effects [28].

### 2.3. Data Analysis

A descriptive analysis of sociodemographics and the acceptability of the sampling methods was performed. A logistic and linear estimation of generalized equations was conducted to determine the association of the acceptability categories and to adjust for the measuring of the three sampling methods.

For this purpose, the category of device and technique acceptability was dichotomized into better when the participants responded with 4: satisfied or 5: very satisfied, and worse when they responded with 1: very dissatisfied, 2: dissatisfied, or 3: indifferent. In the case of the individual perception of the test, those who responded with 1: very low level or 2: low level were classified as better, and those who responded with 3: intermediate level, 4: high level, or 5: very high level were classified as worse. In the category of an overall preference of the method, participants were considered better when they responded with 3: I would always choose (very satisfied), and very dissatisfied when they responded with 1: I would not choose it again or 2: indifferent.

Completed questionnaires with sociodemographic data and the results of HPV tests were transcribed to a Microsoft Excel 2016 spreadsheet for cleaning and coding and were subsequently transferred to R version 4.2.0. Descriptive statistics were presented using means and standard deviation (SD) for continuous variables, while frequencies and percentages were used for categorical variables.

## 3. Results

A total of 120 women participated in this study, all of them living in the rural parish. The sociodemographic characteristics of the participants are shown in Table 1.

Figure 1 shows the acceptability of the device and technique among the three tests. The participants considered urine sampling faster to use, 107 (89.2%); more comfortable, 112 (93.3%); and easier to use, 115 (95.8%). Of the women, 117 (97.5%) considered urine and vaginal self-sampling as granting equal privacy. In terms of confidence in taking the sample, 90 (75.0%) were equally very satisfied with clinician sampling and vaginal self-sampling. However, 96 (80.8%) believed that clinician sampling is more reliable for a diagnosis compared to the other two methods.

Figure 2 shows the individual perception of acceptability. The participants considered urine less painful 117 (97.5%); less difficult to sampling obtention 115 (95.85%) Both methods, and vaginal self-sampling have the lowest level of embarrassment 115 (95.8%). However, clinician sampling has the lower level of mistrust among the methods 97 (80.8%)

Figure 3 presents the overall preference among the methods. Of the participants, 101 (84.2%) will always choose vaginal self-sampling, followed by urine sampling, 66 (55.05%), and clinician sampling, 62 (51.7%). It is remarkable that 38 (31.7%) participants would never choose clinician sampling.

## 4. Discussion

The aim of this study was to compare the acceptability of urine and vaginal self-sampling methods versus clinician sampling among rural women in Ecuador.

Both self-sampling methods were reported to be more comfortable, more private, and less painful as well as embarrassing compared with clinician sampling. Similar results were reported by Nelson et al.; a systematic review that included 18,516 women found that participants considered vaginal self-sampling easy to use, 96%: more private, 95%; less embarrassing, 97%; more comfortable, 96%; and less painful, 98%, compared with clinician sampling [18]. Sultana et al. reported similar results for vaginal self-sampling in a cohort of 573 participants: easier to use, 93%; less embarrassing, 94.3%; and more convenient, 97.7%, compared with traditional sampling. However, 57.4% of all of the participants were unsure about the reliability of the test compared with the clinician sampling [29]. The values of reliability are comparable with the results presented by Shin et al., where the trust in the results of vaginal self-sampling reached 87.5% [17]. This variation could be explained by information to the participants before the sampling test.

Urine self-sampling is considered slightly faster, more private, less embarrassing, less painful, easy to use, and less difficult to use compared with vaginal self-sampling. Similar results were reported by Shin et al.; in this study, urine sampling was considered less embarrassing, 92.9%; less painful, 97.5%; causes less anxiety, 94.2%; and causes less discomfort, 85.9%, compared with vaginal sampling. However, trust in the reliability of this test was inferior to the report of Shin et al.: 91.4% versus 66% [17]. An explanation of this difference was provided by some comments of participants in regard to worries about the logistical chain of the transportation of the sample and the belief that a simple urine test is not able to detect viruses such as HPV.

Our results show that vaginal self-sampling (OR 4.97) had a higher overall preference over urine sampling (OR 1.14) This result differs from the report of Shin et al., where urine self-sampling had a slightly higher preference compared with vaginal self-sampling, OR 2.47 versus 2.09 [17]. An explanation for this could be that some participants consider the latter as less reliable, although this was not a finding in our sample; some mistrust about the quality of a sample could decrease satisfaction in this test [30]. In addition, women in rural communities or ethnic minorities prefer clinician sampling because they believe in the reliability of this sample [17,31].

Self-sampling methods have demonstrated their sensitivity and specificity for the diagnosis of HPV with different sampling devices, including urine collection [32]. Since these methods are not broadly applied, women and health professionals are not familiar with the benefits and effective properties of these tests. Doubts related to self-capability in their application are present among women and health professionals [33].

However, since this is the first time that women have faced self-sampling methods in Ecuador, all of the participants were willing to carry out self-sampling collections after an explanation with graphics and the devices used. Training and teaching women how to perform self-sampling increases their confidence to perform these tests, and women consider them easy to use when following clear instructions [18,34].

HPV self-sampling has high acceptability among women; home-based HPV screening has high acceptability as well [29]. A literature review has shown that offering self-sampling methods for HPV screening at home by delivering devices by mail could save time and also increase the adherence to and coverage of cancer prevention [33].

Due to this high acceptance, these methods could be useful to expand the coverage of cervical cancer screening and also to enroll underscreened or never-screened women, which are named hard-to-reach women [18,25].

Self-sampling methods could substitute gynecological examinations in places with a lack of health infrastructure, being cost-effective and well-accepted options, in order to increase cancer screening uptake [26,35]. Moreover, since the lockdown due to COVID-19 has decreased cervical cancer screening, self-sampling methods could be part of the solution for similar situations [36,37].

### Limitations

Several limitations of this study were identified: the first one is related to the limited number of participants. However, since this is the first study on self-sampling methods in Ecuador, the opinions of the participants could be free of any bias or prejudice about these tests, and, in addition, since the three tests were tested on the same day with the same group of women, their opinion could be more trustworthy. A second limitation is that the sample was selected for convenience.

Therefore, a limited number of participants was recruited. Further research will allow the inclusion of more women in order to evaluate, to a greater extent, the perceptions about the self-sampling collection.

## 5. Conclusions

This research study was the first one conducted in Ecuador that evaluated the acceptability of self-sampling methods in a rural community in Cuenca, Ecuador.

The acceptability of vaginal and urine self-sampling methods is higher compared with clinician sampling. Participants consider self-collected techniques faster, more comfortable, easier to use, and more private. Additionally, women feel, after the use of the three methods, that urine and vaginal self-sampling are less painful and embarrassing, as well as easy to use. However, women still struggle with the self-capability of correctly taking a sample as well as the reliability of a diagnosis of a self-sampling test.

The overall preference of sampling methods allocates vaginal self-sampling in the first place, followed by urine self-sampling and clinician sampling.

Self-sampling methods have high acceptability in a rural context; this observation, together with the good diagnostic accuracy of these methods, could increase the rates of cervical cancer screening uptake. However, previous training should be carried out at the community level to obtain a better application of self-testing and to counter misconceptions about the reliability of self-sampling methods.

## Figures and Tables

**Figure 1 healthcare-10-01614-f001:**
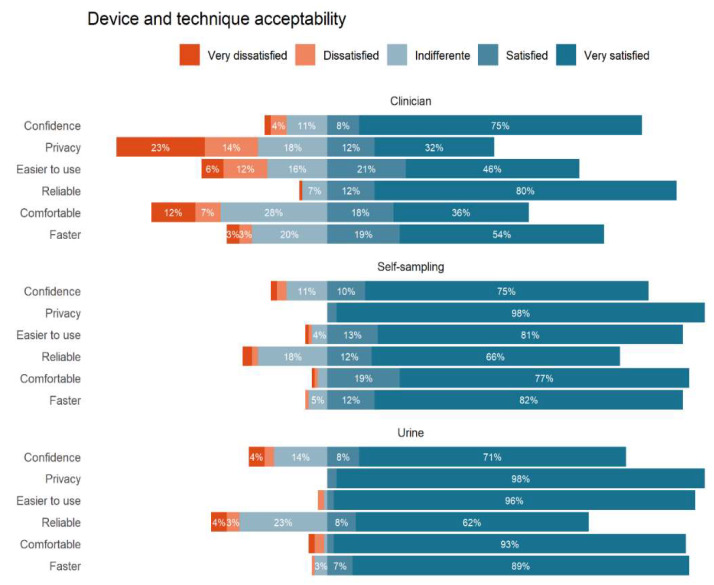
Device and technique acceptability.

**Figure 2 healthcare-10-01614-f002:**
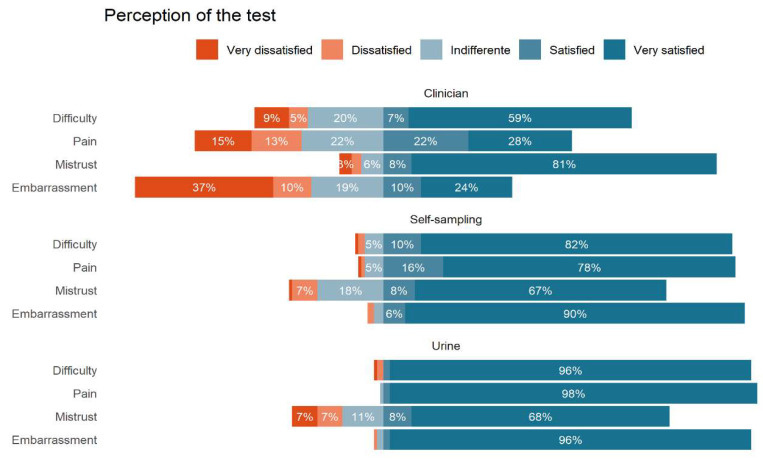
Individual perception of acceptability.

**Figure 3 healthcare-10-01614-f003:**
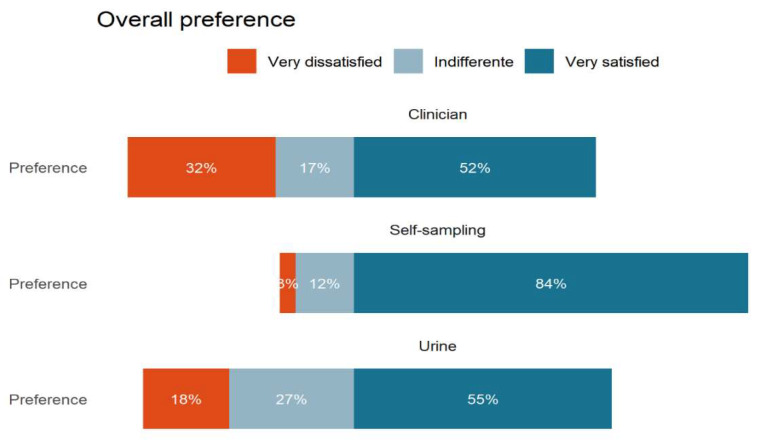
Overall preference of the sampling methods.

**Table 1 healthcare-10-01614-t001:** Participants’ sociodemographic characteristics.

Variable		N (%)
**Age**	Age median: 35; mode: 24; and SD ± 11.23	
	<35 years	60 (50%)
	>35 years	60 (50%)
**Level of instruction**		
	None or primary	65 (54.17%)
	Secondary or superior	55 (45.83%)
**Civil status**		
	Married or stable union	77 (64.17%)
	Divorced, widow	18 (15%)
	Single	25 (20.83%)
**Family** **income per month**		
	<USD 100	22 (18.33%)
	USD 101 to 200	44 (36.67%)
	USD 201 to 400	19 (15.83%)
	>USD 400	35 (29.17%)
**Occupation**		
	Housewife	69 (57.5%)
	Merchant	13 (10.83%)
	Employed	27 (22.5%)
	Others	11 (9.17%)
**Currently working**		
	Yes	45 (37.5%)
	No	75 (62.5%)
**Number of Pap tests during lifetime**	Median: 5.24; mode: 3; minimum: 0; maximum: 3; and SD ± 11.23	

## Data Availability

The datasets generated and/or analyzed during the current study are not publicly available because they contain the sensitive personal information of the participants. The informed consent grants the confidentiality of the participant’s data. However, the datasets are available from the corresponding author upon reasonable request.

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
