# Peer review of "Evaluation of Urine and Vaginal Self-Sampling versus Clinician-Based Sampling for Cervical Cancer Screening: A Field Comparison of the Acceptability of Three Sampling Tests in a Rural Community of Cuenca, Ecuador"

_healthcare, 2022, doi:10.3390/healthcare10091614_

Round 1

Reviewer 1 Report

Dear author's
I was pleased to review your manuscript. The subject is not new, this type of screening is already used in some countries. 
I have the following comments:
- Introduction: The introduction is focused in detection of CC and not in the acceptability or reliability of self-sampling methods. Please resume the information about CC in Latin America and add the incidence of this cancer and the mortality rates.

Methods

- The informed consent was obtained form the patients?

- The sample of the study is relatively small.

Discussion

In this section it is mandatory to compare your results with the international literature.

Knowing this results do you think that the screening can be extended in this part of Latin America?

Please specify the novelty of your study.

Author Response

Dear Reviewer

Please find below our responses

Thank you so much for taking the time to review our work

Dear author's

I was pleased to review your manuscript. The subject is not new, this type of screening is already used in some countries. 

Response

Thank you so much we do really appreciate your time 

I have the following comments:

1.- Question

- Introduction: The introduction is focused on detection of CC and not in the acceptability or reliability of self-sampling methods. Please resume the information about CC in Latin America and add the incidence of this cancer and the mortality rates.

Response

Thank you so much for your suggestions, we have added some paragraphs about acceptability and reliability of self-sampling methods and resume the information about cervical cancer and ad incidence rates of mortality.

Methods

2 Question

- The informed consent was obtained from the patients?

Response

Yes, every woman that underwent to urine self- sampling, vaginal self-sampling and clinician sampling signed and informed consent previous the sampling taking.

For a better understanding, we changed patients to women or participants

3 Question

- The sample of the study is relatively small.

Response

This study is part of a diagnostic test to compare sensitivity, specificity and predictive values of urine, vaginal self-sampling compared with clinician sampling. A prepositive sampling was made to reach a enough number of participants and have a balance with the funding of the project.

We have added that is a limitation for this study is a relatively small number of participants.

Discussion

4.- Question

In this section it is mandatory to compare your results with the international literature.

We have added a comparison with international literature in the discussion

5.- Question

 Knowing this results do you think that the screening can be extended in this part of Latin America?

Response

We hope that the results will be useful for extend self-sampling in Ecuador and in Latin America, this article is part of one published about sensitivity and specificity about self-sampling https://www.mdpi.com/1660-4601/19/8/4619/htm

We are generating local evidence in terms of acceptability and sensitivity of self-sampling methods, therefore we had presented our results to PAHO and the Ministry of Health in Ecuador

6.- Question

Please specify the novelty of your study.

As you mentioned this type of screening is used in some countries, most of them have high rates of coverage of CC screening. This initiative is applied for first time in Ecuador, and is showing good results in terms of acceptability, specificity and sensitivity. The idea with this local validation is to ensure previous steps the share the knowledge with governmental institutions and reach a high rate under screened women.

Since is the first time that those methods are applied in Ecuador, all is novel for clinicians and also for the women at the community, that face for first time self-sampling, since the acceptability is adequate this could give a pathway in order to distribute self-sampling test through traditional healers and in places where infrastructure doesn’t allow a clinician sampling.

The first step to implement self-sampling methods at rural communities is to understand in what extend will they accept these relative new techniques in Ecuador    

7.- For a better presentation of the document we have corrected English language editing and correction

Reviewer 2 Report

Suggestions:

Line 25: methods for HPV testing

LIne 27: change could increase to have been shown to increase

LIne 41: Conclusions:

LIne 48: first cancer ELIMINATED (WHO has a stong definition between elimination and eradication and so this must be use appropriately)

LIne 63: main associations rather than causes. Line 64: remove that LIne 65: as instrusive

LIne 79: HPV testing

LIne 99: Why have you included the ages of 18 for your study? HPV testing at it's earliest should not be less than25 years of age. Is this the local guideline?

LIne 116, Finally, the patient was subjected to a clinician acquired sample in the standard manner.

Line117: HybriBio Registred? Copyright? please use the appropriate symbol beside it.

Line 121: were included in the study

For the measurement of the acceptability, did the team use a validated questionnaire?  Have the questionnaire been used elsewhere?  This is an important point. What language was the questionnaire carried out in? If this is a new questionnaire, then a formal validation process is important.

In table 1: Family income per month.

LIne 203: among rural women in Equador. LIne 204: Both self-sampling methods are reported to be more......

LIne 214: consider the latter as less

It is really interesting to note that while urine sampling seems to score well, women still preferred vaginal self sampling.  Can you provide more potential explanation for this?  Will the team be publishing the data on the test performance? I would be really good to have data on the performance of the HPV tests using the different methods i.e. invalid rates.

Author Response

Dear Reviewer

Please find below our responses

Thank you so much for taking the time to review our work

Question 1

Line 25: methods for HPV testing

Response

Thank you so much, we have corrected

Question 2

LIne 27: change could increase to have been shown to increase

Response

Thank you so much, we have corrected

Question 3

LIne 41: Conclusions:

Response

Thank you so much, we have corrected

Question 4

LIne 48: first cancer ELIMINATED (WHO has a stong definition between elimination and eradication and so this must be use appropriately)

Thank you so much, we have changed to eliminated, we have added the citation of the revied paper

Question 5

LIne 63: main associations rather than causes. Line 64: remove that LIne 65: as instrusive

Response

Thank you so much, we have corrected

Question 6

LIne 79: HPV testing

Response

Thank you so much, we have corrected

Question 7

LIne 99: Why have you included the ages of 18 for your study? HPV testing at it's earliest should not be less than25 years of age. Is this the local guideline?

Response

Yes, indeed, local guidelines also recommend include women up to 25 years of HPV testing. However due the lowest rates of coverage, the early sexual onset in Ecuador (average 15 years), and due a small rate of women that has been diagnosed or dead by cervical cancer bellow 25, national protocols, do not exclude women bellow 25 that attending for screening.

However, the main reason is that the study of acceptability is part of the study of sensitivity of self-sampling methods. In order to have enough number of positive HPV samples we included women up from 18 years old

Question 8

LIne 116, Finally, the patient was subjected to a clinician acquired sample in the standard manner.

Response

Thank you so much, we have corrected

Question 9

Line117: HybriBio Registred? Copyright? please use the appropriate symbol beside it.

Response

Thank you so much, we have corrected

Question 10

Line 121: were included in the study

Response

Thank you so much, we have corrected

Question 10

For the measurement of the acceptability, did the team use a validated questionnaire?  Have the questionnaire been used elsewhere?  This is an important point. What language was the questionnaire carried out in? If this is a new questionnaire, then a formal validation process is important.

Response

The questionnaire of acceptability has been used in Korea in 2019 by Shin et al. Similar formulary was used by Sultana in Australia 2015.  The questionnaire for acceptability was translated from English to Spanish and validated his content in Spanish by experts of the team.

We have added this reference to the document.  

Question 11

In table 1: Family income per month.

Response

Thank you so much, we have corrected

Question 12

LIne 203: among rural women in Equador

Response

Thank you so much, we have corrected

Question 13

LIne 204: Both self-sampling methods are reported to be more......

Response

Thank you so much, we have corrected

Question 14

LIne 214: consider the latter as less

Response

Thank you so much, we have corrected

Question 15

It is really interesting to note that while urine sampling seems to score well, women still preferred vaginal self sampling.  Can you provide more potential explanation for this? 

Response

An explanation of this difference was provided by some participants about worries in the logistic chain of transportation of the sample and the beliefs that a simple urine test is not able to detect virus like HPV

They also manifested their worries about contamination of the sample, spill of the sample during the transportation and also contamination  during sampling collections 

Question 16

Will the team be publishing the data on the test performance? I would be really good to have data on the performance of the HPV tests using the different methods i.e. invalid rates.

Indeed, we have published an article about the performance of methods

Role of Self-Sampling for Cervical Cancer Screening: Diagnostic Test Properties of Three Tests for the Diagnosis of HPV in Rural Communities of Cuenca, Ecuador

https://www.mdpi.com/1660-4601/19/8/4619

17.- For a better presentation of the document we have corrected English language editing and correction